# Sustainable elk harvests in Alberta with increasing predator populations

**Tyler Trump**[1], **Kyle Knopff**[1,2], **Andrea Morehouse**[1,3], **Mark S. Boyce**[1]*

**1** Department of Biological Sciences, University of Alberta, Edmonton, Alberta, Canada, **2** Golder Associates, Calgary, Alberta, Canada, **3** Winisk Research and Consulting, Pincher Creek, Alberta, Canada

* boyce@ualberta.ca

## Abstract

Large predators often are believed to cause declines in hunter harvests of ungulates due to direct competition for prey with hunters. In Alberta, predators of elk (*Cervus elaphus*), including grizzly bear (*Ursus arctos*), cougar (*Puma concolor*), and wolf (*Canis lupus*), have increased in recent years. We used trend analysis replicated by Wildlife Management Unit (WMU) to examine regional trends in elk harvest and hunter success. Over a 26-yr period, average harvest of elk increased by 5.46% per year for unrestricted bull and by 6.64% per year for limited-quota seasons. Also, over the same time frame, average hunter success increased by 0.2% per year for unrestricted bull and by 0.3% per year for limited-quota seasons, but no trend was detected in hunter effort (P>0.05). Our results show that increasing large-predator populations do not necessarily reduce hunter harvest of elk, and we only found evidence for this in Alberta's mountain WMUs where predation on elk calves has reduced recruitment. Furthermore, data indicate that Alberta's elk harvest management has been sustainable, i.e., hunting has continued while populations of elk have increased throughout most of the province. Wildlife agencies can justify commitments to long-term population monitoring because data allow adaptive management and can inform stakeholders on the status of populations.

## Introduction

Elk (*Cervus elaphus*) are an important big game species in Alberta, Canada. After being nearly extirpated from the province 100 years ago, elk populations have been restored through translocations and harvest management. Despite their importance, elk populations are infrequently monitored. Like several other jurisdictions in western North America, elk population monitoring in Alberta has been done predominately by aerial surveys [1]. Because the cost of aerial monitoring is high, aerial surveys in Alberta are conducted infrequently, typically only once every 10 years [2]. Nearly 80% of wildlife agencies across Canada and the United States collect data on harvest [3]. However, in many jurisdictions these data are insufficiently analyzed to permit agencies to evaluate the efficacy of their management [4].

A common objective for elk management is to ensure sustainable hunter harvests where continued harvest does not result in population declines. With few data available for setting

**Data Availability Statement:** Data are available in dataverse managed by the University of Alberta library. Now posted with doi.org/10.5683/SP3/DINHMO.

**Funding:** This work was supported by the Alberta Conservation Association (ACA, Grant number RES0034583) https://www.ab-conservation.com/ received by MSB; the Alberta Fish and Game Association (AFGA, Grant number RES0035258) https://www.afga.org/ received by MSB; and Safari Club International Northern Alberta Chapter (SCINAC, Grant number RES0034875) http://sci-northern.ab.ca/ received by MSB. The funders had no role in study design, data collection and analysis, decision to publish, or preparation of the manuscript.

**Competing interests:** The authors have declared that no competing interests exist.

regulations and quotas, harvest management strategies change little unless something appears to have gone wrong, e.g., a sharp decline in harvests or anecdotal field reports by biologists and hunters. Elk harvests in Alberta are mostly regulated by harvests under general or limited-quota licenses. General harvests, also known as open-entry harvests, do not limit the number of resident hunters who can hold this license type, but they are controlled with antler-point restrictions (APRs) that target specific age and sex classes [5]. In many Alberta WMUs elk hunters have a 3-point minimum (elk having an antler that has two tines that are 3 inches or greater projecting from a main beam) and in a few WMUs 6-point minimum (one antler must have at least five tines 3 inches or greater projecting from a main beam) APRs. General harvests with APR's can limit survival of bulls to older age classes [6, 7], but are thought to offer maximum hunter yields while protecting reproductively significant cows and breeding-capable subadult males [8]. Limited-quota harvests, also known as special or limited-entry harvests, restrict the number of hunters who can participate by limiting the number of licenses to achieve a harvest quota. Licenses are distributed by random draw of applicants. By limiting licenses sold, limited-quota can limit hunter harvests, but by allowing designation of females and calves as well as males, these licenses offer wildlife managers better control over the elk population than with general harvests of branch-antlered males. Other jurisdictions in North America usually have similar license restrictions although details vary.

Due to recent conservation efforts, large carnivore populations have been recovering in many portions of both North America and Europe, attributable to increasing human tolerance [9] and increases in ungulate prey [10]. Increases in ungulate abundance have resulted in part from legislation that aims to ensure sustainable harvests by hunting, and successes in science-based management [3]. These statements hold true in Alberta, where grizzly bear (*Ursus arctos*, [11]), cougar (*Puma concolor*, [12]), and wolf (*Canis lupus*, [13]) populations have been increasing, as have damage claims on livestock depredation [14]. With these population increases, a common belief about large predators is that they compete with hunters by decreasing ungulate populations through additive mortality [15–19], thereby resulting in decreased hunter harvest and hunter success.

Societal goals in the form of hunter satisfaction often accompany biological goals of a wildlife agency [20]. Aggregate hunter satisfaction can be difficult to measure because what one hunter views as a satisfactory hunt might be different for another hunter. For example, hunter age and lifetime hunting experience [21], hunter to hunter interaction and viewing harvestable wildlife [22], trophy characteristics [20, 23], and species of the hunted animal [21], can influence perception of a satisfactory hunt. Quantifiable measures of satisfaction commonly collected by wildlife agencies include hunter success and hunter effort [24, 25], with success being defined as a kill of the target species and hunter effort defined as the number of days spent hunting.

Alberta has collected hunter harvest and success data for elk but has not evaluated the results of regulations or trends, particularly in context of growing predator populations. Therefore, our objective was to assess the results of Alberta's hunter harvest, hunter success, and hunter effort in relation to the increasing predator populations within the province. We envisage two questions that can be answered from an analysis of these hunter-harvest data: (1) has harvest management been sustainable? and (2) have elk harvests declined because of increasing large predator populations? To evaluate the trend in hunter harvest and hunter success, we examined harvest data from 1995 to 2020 collected by Alberta Environment and Parks (AEP) [26]. Because of increases in the populations of all three of Alberta's large predators, we expected to find a declining trend in total harvest and hunter success.

## Study area

For purposes of wildlife management, the province of Alberta is divided into Wildlife Management Units (WMU), legislatively recognized areas of land for which harvest regulations are designated. There are currently 189 WMUs in Alberta and 148 of those have regulated elk harvests. In addition to elk, WMUs are used to distribute and monitor hunter harvests for moose (*Alces alces*), mule deer (*Odocoileus hemionus*), white-tailed deer (*O. virginianus*), bighorns (*Ovis canadensis*), and pronghorn (*Antilocapra americana*). WMUs throughout the province have gone through many border adjustments over time, resulting in more WMUs currently than in the past. However, during the time frame of our study (1995–2020) WMUs have remained mostly constant. WMUs can be grouped into larger Zones that coarsely mimic natural ecological regions and sub-regions of Alberta [27]. These 5 zones include the Prairie (Zone 1), Parkland (Zone 2), Foothills (Zone 3), Mountain (Zone 4), and Northern Boreal WMU's (Zone 5) (Table 1). Hunting is prohibited in Jasper, Banff, Waterton Lakes, and Wood Buffalo National Parks as well as most provincial parks and recreation areas. Areas with no licensed hunter harvests were excluded from our analysis.

## Methods

### Large carnivore abundance

We used data from government reports and previously published studies of large carnivore populations in Alberta to document changes in abundance and distribution. We inferred cougar and wolf population growth in Alberta using provincial human-caused mortality data for cougars during 1971–2010 [12] and trapping data for wolves during 1985–2006 [13]. We reviewed provincial records and the literature for estimates of Alberta grizzly bear abundance during the period of this study (1999–2016). Species status assessments for grizzly bears were published in 2002 and 2010 [28, 29] and an updated recovery plan in 2021 [30]; these

**Table 1. Alberta's 5 zones separated by natural region, defining characteristics, and total elk harvest and hunter success.** For a more detailed description of each Zone, use the Natural Regions and Subregions of Alberta (Natural Regions Committee 2006).

| Zone and WMU's | Natural Region/km$^2$ | Defining Characteristics | Total Harvest (H) and Annual Hunter Success (S) by Harvest type | |
| --- | --- | --- | --- | --- |
| | | | General 1995–2020 | Limited quota 1995–2019 |
| **Zone 1: Prairie WMU's**<br>• 100's<br>• 732 | Grassland Natural Region<br>• 95,565 km$^2$ | • Level plains and rolling hills<br>• Mixed grasses<br>• Few rivers and lakes | • H: 401<br>• S: 10.32% | • H: 7,594<br>• S: 49.26% |
| **Zone 2: Parkland WMU's**<br>• 200's<br>• 728,730, 936 | Parkland Natural Region<br>• 60,747 km$^2$ | • Rolling hills<br>• Grasslands and aspen stands<br>• Mostly cultivated | • H: 6,690<br>• S: 9.30% | • H: 5,968<br>• S: 32.80% |
| **Zone 3: Foothills WMU's**<br>• 300's | Foothills Natural Region<br>• 66,436 km$^2$ | • Rolling hills to mountainous<br>• Mixed forests | • H: 39,336<br>• S: 7.76% | • H: 34,810<br>• S: 34.06% |
| **Zone 4: Mountain WMU's**<br>• 400's | Rocky Mountain Natural Region<br>• 49,070 km$^2$ | • Mountainous, deep valleys, elevated meadows<br>• Mixed forests, open grasslands, barren mountain tops | • H: 4,456<br>• S: 4.18% | • H: 2,983<br>• S: 22.13% |
| **Zone 5: Northern Boreal WMU's**<br>• 500's<br>• 841 | Boreal Forest Natural Region<br>• 381,046 km$^2$<br>Canadian Shield Natural Regions<br>• 9,719 km$^2$ | Boreal Forest<br>• Flat plains and rolling hills<br>• Mixed forests<br>• Numerous wetlands<br>Canadian Shield<br>• Rolling hills of exposed bedrock<br>• Forests where possible<br>• Lichens, mosses, and ferns | • H: 10,807<br>• S: 11.06% | • H: 13,171<br>• S: 27.00% |

documents provide information on the overall density, distribution, and abundance of grizzly bears in the province (Fig 1).

## Harvest estimates

We obtained data on estimated elk harvests from 1995–2020 from AEP [26]. All estimates were based on hunter responses to harvest surveys that were delivered post-harvest to people who bought a hunting license, although survey methods varied among years. From 1995 to the early 2000s surveys were delivered to hunters by post or by telephone. In the mid to late 2000s, AEP shifted to a combination of email and mail-in surveys that have persisted past 2017. No harvest estimates were available prior to 1995. Hunters were encouraged, but not required, to complete post-harvest surveys resulting in a degree of non-response. AEP has accounted for this non-response by using data from hunters who did respond and extrapolating to the remaining hunter population. This assumes that the proportion of harvest success among hunters who responded is the same as those who did not respond and that the surveys are representative of Alberta's actual hunter harvest and success. Even if a bias exists because of this assumption so long as it remained roughly equivalent over time the assumption would have little consequence to our analysis. Harvest surveys also were used to obtain a record of the number of days that each hunter spent hunting. Surveys also provided data on whether the hunt was successful or not, and if the hunt was successful, data were collected on the class of animal harvested (e.g., bull, cow, or juvenile).

## Trend estimates

We digitized the history of hunting regulations 1970–2020 for each WMU, as well as beginning and end dates for each harvest season. We compiled the estimated elk harvest and hunter success for each WMU from the harvest surveys between 1995–2020 [26] to link elk harvested with the respective general and limited-quota regulations. Lastly, we applied the respective Zone designation (1–5) to each WMU.

We used linear regression of harvest vs time to estimate trends in harvest and Spearman rank to assess trend in hunter harvest and success across time for both general and limited-quota harvests. Trends for individual WMUs would be temporally autocorrelated for each of these relations, thus we used a method similar to route regression [31, 32], where replication within a zone was obtained by an analysis of slopes by WMU. Average slopes can then be compared to an expectation of no change, i.e., zero slope, or comparisons can be made using a t-test [31].

# Results

## Large carnivore abundance

Mortality data for cougars clearly demonstrate range expansion in Alberta out of the mountains and into other natural regions [12] and strongly suggest that populations have increased after 1970 when systematic persecution had reduced cougars to low levels.

Similarly, mortality data for wolves indicate a population increase between 1995 and 2006 [13], following total extirpation in southern Alberta for rabies control in the 1950s when >4,200 wolves were killed mostly with toxicants [33]. After 30 years without wolves, they returned to Banff National Park in 1985 and quickly recolonized the Rocky Mountains into Montana [13, 33].

Although mortality trends suggest an increase, the extent to which cougar and wolf populations grew during 1995–2016 is difficult to determine and we caution that human-caused

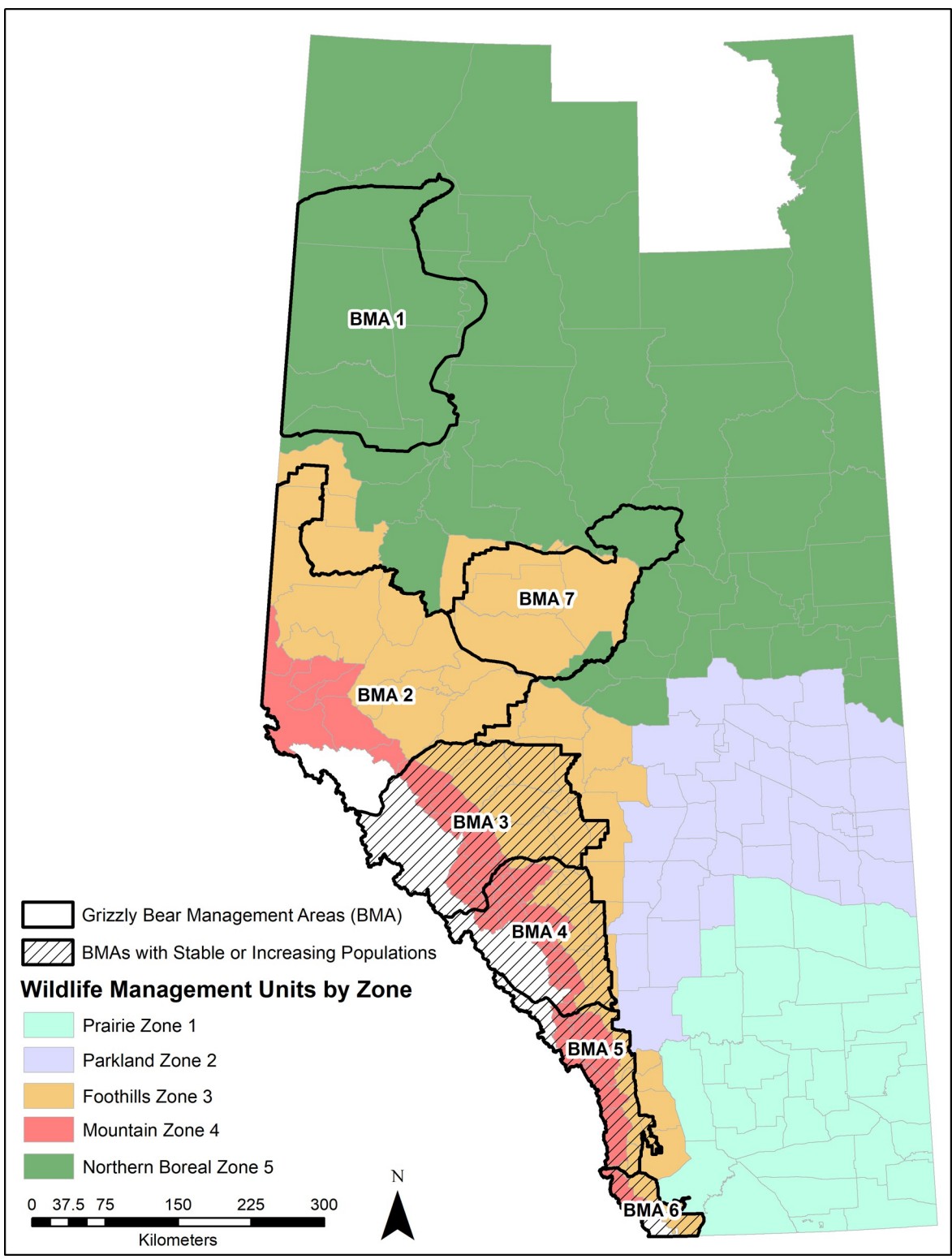

**Fig 1. Wildlife management units of Alberta, Canada by wildlife management area/zone.** Overlaid are Alberta's seven grizzly bear management areas (BMAs). BMAs with stable or increasing grizzly bear populations are denoted by black hatched fill. Map by A. Morehouse.

mortality data also might change due to increased harvest effort [12, 13]. In the case of cougars, substantial increases in combined hunting and non-hunting sources of human-caused mortality despite declining harvest quotas during 2000–2010 [12] strongly indicate cougar population growth during the period over which we monitored elk harvest. Although we were able to infer an increase in populations of cougars and wolves during the period of our study, the magnitude of increase could not be estimated.

In 2000, the estimated provincial grizzly bear population (excluding bears in national parks) was estimated to be 841 [28]. Biologists estimated between 175 and 185 bears in Alberta's national parks, bringing the total 2000 provincial estimate to between 1,016 and 1026 grizzly bears [28]. This number represented an increasing provincial trend since the late 1980s [28]. The next provincial estimate was released in 2010 and was based on a series of DNA-based population inventories [29]. The 2010 grizzly bear status assessment estimated 691 grizzly bears in Alberta plus additional bears in portions of Banff and Jasper National Parks [29]. The most current DNA-based provincial estimates were released in early 2021 and indicate there are >750 grizzly bears outside national parks in Alberta [30]. Provincial estimates indicate a stable or increasing population trend [30]. In particular, between 2008 and 2018 Bear Management Areas (BMAs) 3 and 4 have had large increases in grizzly bear abundance–annual population rate of increase of 7% and 6% respectively [34, 50]. These BMAs are largely in the Mountain Zone 4 WMUs.

## Regulations

Before 1973, regulations in Alberta allowed harvest of both antlered and antlerless elk during general seasons [35]. Between 1973 and 1987 the first antler point-based system, a 5-point antler minimum general season, was introduced and was replaced in 1988 with either a 6-point or a 3-point resident/6-point nonresident general season. Over the next few years, all WMUs independently lost the resident and nonresident general harvest designations and all WMUs with general seasons had 6-point or 3-point APRs. To limit the female elk harvest in 1975, the antlerless general season became either an archery-only general season or a limited-quota season and has remained that way since.

## Harvest: Temporal and spatial

During our study period, 126,215 elk were harvested in Alberta during general and limited-quota seasons (Table 2). While the two types of hunting seasons resulted in similar harvest numbers of elk, approximately 62,000 for general and 64,000 for limited-quota, the composition of harvest under each regulation type was different, with general-season harvests being primarily bulls and limited-quota harvests being primarily cows and calves.

The number of elk harvested provincially, for both general and limited-quota seasons, has trended upwards indicating that harvests were sustainable (Fig 2). The average harvest in general seasons increased by 5.46% annually, with a ranked correlation between harvest and year, $r_s = 0.70$. Harvests in limited-quota hunts increased by 6.64% annually, with a very high ranked correlation between harvest and year, $r_s = 0.94$.

**Table 2. Total number of elk harvested 1995–2020 for general licenses in Alberta, Canada, and for limited-quota licenses for 1995–2019.**

| Regulation | Bulls | Cows | Juveniles | Total Elk / Regulation |
|---|---|---|---|---|
| General (including General Archery) | 56,704 (92%) | 4312 (7%) | 674 (1%) | 61,690 (100%) |
| Limited quota (including Special Archery) | 6,220 (10%) | 51,070 (79%) | 7,235 (11%) | 64,525 (100%) |
| Total Elk / Class | 62,924 | 55,382 | 7,909 | 126,215 |

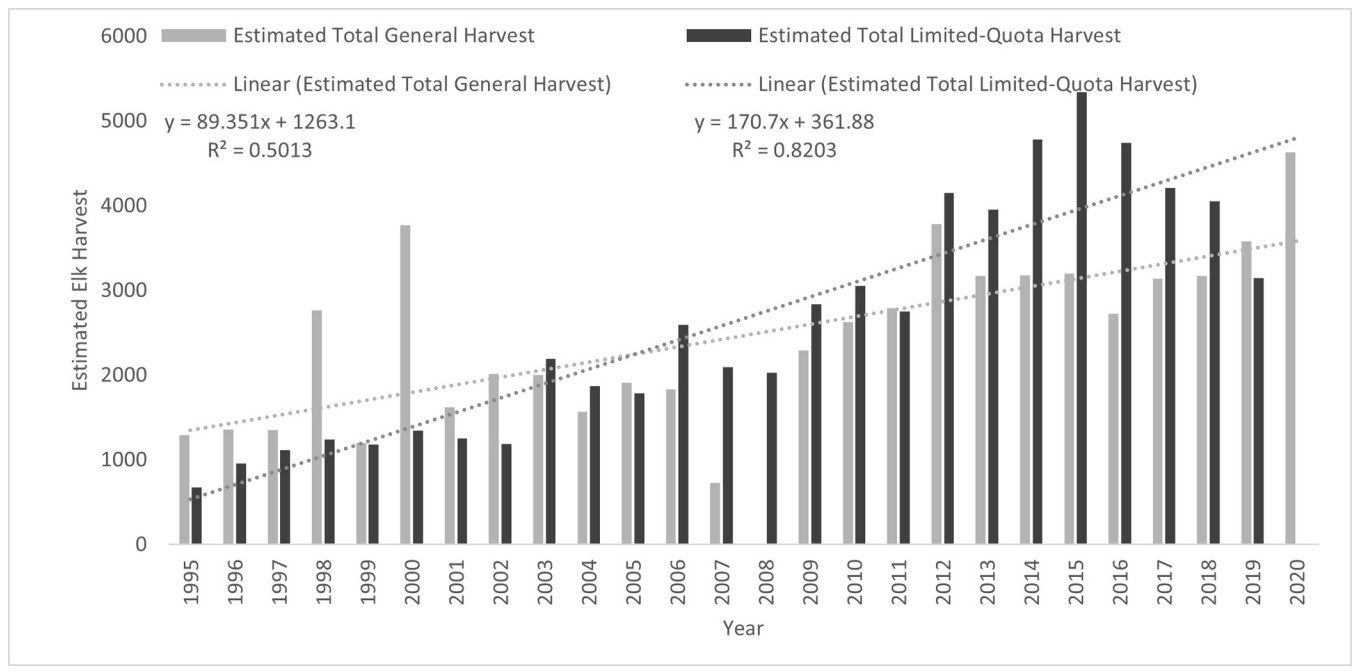

**Fig 2. Total estimated elk harvest in Alberta by year for general and limited-quota harvests from 1995 to 2020 across all wildlife management units.**

Across all years of analysis, most of the elk were harvested in the foothills and boreal (zones 3 and 5) with 39,336 (63.76%) and 10,807 (17.52%) elk taken respectively during general seasons and 34,810 (53.95%) and 13,171 (20.41%) elk, respectively during limited-quota seasons (Table 1). Zones 1, 2, and 4 accounted for 401 (0.65%); 6,690 (10.85%); and 4,456 (7.22%) elk respectively in the general elk harvest, while Zones 1, 2, and 4 accounted for 7,594 (11.77%); 5,968 (9.25%); and 2,983 (4.62%) elk respectively during limited-quota seasons.

## Hunter success and effort: Temporal and spatial

The mean annual hunter success rate was 9.2% during general seasons and 33.5% for limited-quota seasons, each trending upwards over time (Fig 3). General-season hunter success increased by 0.002 annually, with a significant correlation between hunter success and year, $r_s$ = 0.81. For limited-quota seasons, hunter success increased by 0.003 annually, also with a significant ranked correlation between hunter success and year, $r_s$ = 0.51. These trends in hunter success were not attributable to changes in hunter effort because we found no correlation between hunter effort and year ($r_s$ = 0.06, $P > 0.05$; Fig 4).

For the five natural regions, Zones 5 and 1 had the highest mean hunter success for general seasons at 11.1% and 10.3%, respectively, while Zones 2 (9.3%), 3 (7.8%), and 4 (4.2%) had somewhat lower mean hunter success (Tables 1 and 3). Zone 1 had the highest mean hunter success for limited-quota seasons (49.3%). Hereafter, mean hunter success declined for limited-quota seasons in order of Zone 3 (34.1%), 2 (32.8%), 5 (27.0%), and 4 (22.1%).

## Discussion

Although AEP has not evaluated how elk hunter harvest and hunter success has changed in recent years, their harvest policies have been sustainable and have resulted in positive trends in both harvests and hunter success over time [36]. The number of elk hunters also has increased annually since 1995 for both general and limited-quota seasons [26]. With a rise in the number

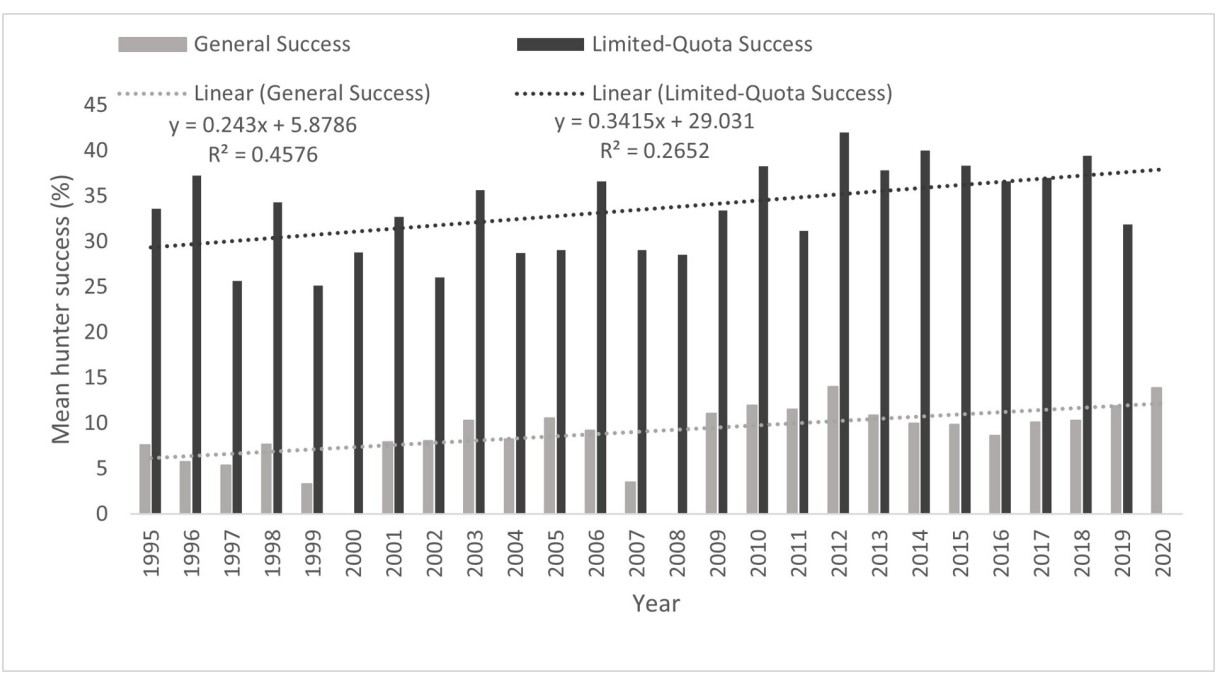

**Fig 3. Mean annual hunter success (%) for general and limited-quota special elk harvests in Alberta from 1995 to 2020 across all wildlife management units.**

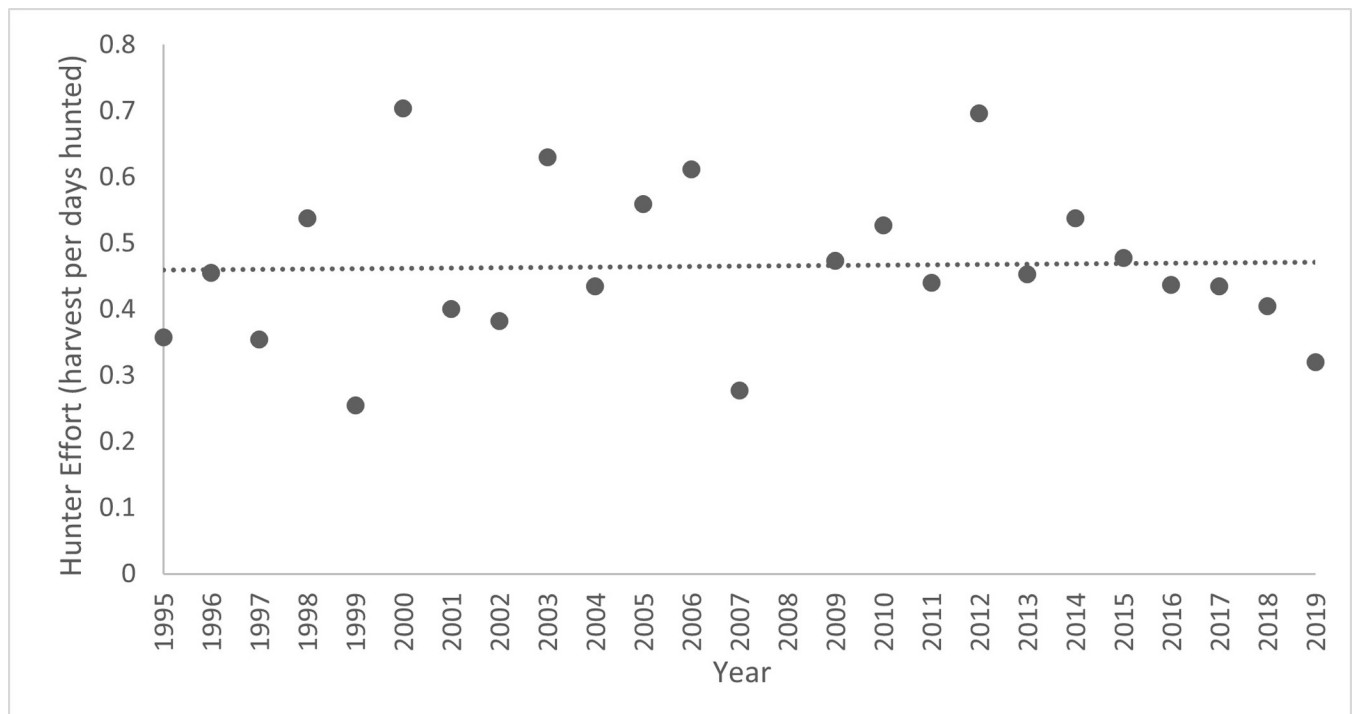

**Fig 4. Annual hunter effort (total harvest per number of days hunted) for Alberta's licensed elk hunters from 1995 to 2020.** We found no temporal trend in hunter effort (P > 0.05).

**Table 3. Slope and standard error of the mean elk hunter harvest and hunter success for WMUs within each zone for general and limited quota licenses in Alberta, Canada from 1995–2020.**

| Harvest Type | Zone | Hunter Harvest | | Hunter Success | |
|---|---|---|---|---|---|
| | | Mean Slope | Standard Error | Mean Slope | Standard Error |
| General (1995–2020) | Zone 1 - Prairie units | 0.01 | 0.52 | 0.01 | 0.02 |
| | Zone 2 - Parkland units | 0.47 | 0.14 | -0.003 | 0.004 |
| | Zone 3 - Foothill units | 1.38 | 0.46 | 0.001 | 0.0004 |
| | Zone 4 - Mountain units | -0.07 | 0.07 | -0.001 | 0.001 |
| | Zone 5 - Northern Boreal units | 1.92 | 0.59 | 0.01 | 0.003 |
| Limited quota (1995–2019) | Zone 1 - Prairie units | 2.30 | 3.96 | 0.01 | 0.03 |
| | Zone 2 - Parkland units | 0.25 | 1.00 | -0.02 | 0.01 |
| | Zone 3 - Foothill units | 1.34 | 0.90 | -0.006 | 0.002 |
| | Zone 4 - Mountain units | -0.49 | 0.11 | -0.004 | 0.002 |
| | Zone 5 - Northern Boreal units | 2.40 | 0.83 | 0.01 | 0.02 |

of hunters from 17,045 in 1995 to 33,355 in 2020 for general-season harvests and 2,003 in 1995 to 9,880 in 2019 for limited-quota harvests [26], an increase in both elk harvested and hunter success, but with no significant change in elk hunter effort, reinforces the data indicating that Alberta's elk are increasing at the provincial scale. The exceptions are only in mountain units where there are the highest concentrations of predators.

From 1995 to 2020, most of the bull harvest was under general license, whereas limited-quota licenses were targeted to harvest mostly antlerless elk. In ungulate herds, bull demographic tends to have relatively little consequence for overall recruitment [2, 37–39]. For example, sex ratios of elk populations can be as skewed as 1 bull for every 25 cows, before reproductive performance is negatively influenced [40, 41]. This allows Alberta to manage its bull elk with APRs, protecting cows and juveniles while still maintaining hunter opportunity [42]. We also found that limited-quota licenses primarily are used by wildlife managers to target females and juveniles [37–39]. These limited-quota licenses are allotted to hunters in limited numbers to keep removals moderate. However, in areas having conflicts with agriculture, antlerless removals can be used to reduce herd size [5, 43, 44].

Surprisingly, continued increases in hunter harvest have been sustained despite increases in large-predator populations. Although both total elk harvested and predator populations are increasing provincially within Alberta, one exception was found in Zones 4 (Table 3) where elk harvests declined during 1995–2020.

We believe that declines in the mountain units (Zone 4) might be attributed to continued disruption of migration routes by roads and industrial development [45–48], and to predation, especially by grizzly bears [49, 50]. Grizzly bear predation on calves has increased in recent years [50], which is attributable to increases in the grizzly bear population in the mountain zones of Alberta [11, 30, 51, 52], thereby reducing elk recruitment [50, 53]. The mountain WMUs are the only units in Alberta where our initial prediction of reduced elk harvest as a result of increasing predator densities was supported and it is these mountain WMUs where combined wolf, cougar, and grizzly bear numbers are highest [11–14].

The ruggedness of terrain and thickness of vegetation reduces hunter access by increasing effort required by the hunter and decreasing the visibility of the prey animal [54, 55], whereas road access can increase densities of hunters [56]. Separating WMU's by natural region allowed us to examine the relationships between landscape and habitat and hunter harvest and success. The landscapes and vegetation among the 5 natural regions vary from mountains to plains and trees to grasslands. As an example of how topography and habitat might affect

hunter success and harvest, the open, grassy-plains habitats of the Prairie Zone (Zone 1) had one of the greatest annual mean hunter success rates for both general and limited-quota seasons, yet still having the lowest total harvest. High hunter success can be explained by high visibility, which limits opportunities for elk to escape [55]. While most of the elk harvest in Zone 1 comes from limited-quota licenses, low numbers also can be explained by the limited vegetation cover and flat terrain, which provide little habitat security leaving few elk left for harvest [54]. The Foothills (Zone 3) is characterized by rolling hills and mixed forests where more elk were harvested than all the other Zones combined. This area provides optimal habitat for elk with a balance of habitat security and forage in the form of forest patches and grasslands, and it encompasses many known wintering areas for Alberta's migratory elk herds that summer in the mountains [45–48].

Long-term monitoring programs by wildlife agencies often are justified for informing stakeholders [57], avoiding conflicts [3], and for evaluating the results of management interventions to improve techniques [58, 59]. This study highlights the importance of evaluating the results of monitoring data such as harvest surveys, despite a paucity of data about population size. Greater detail about trends in abundance could be obtained by increasing the frequency of aerial surveys [1, 2] or by conducting surveys of hunter observations [60–62]. Although aerial surveys of elk in Alberta have been too infrequent to provide adequate monitoring, when combined with trends in harvests distributed among WMUs, clearly Alberta's harvest management is sustainable. Recently, Alberta Environment and Parks has begun mandatory reporting of harvests by hunters ensuring continued harvest data that should be free of any possible non-reporting bias. Despite increasing numbers of elk hunters and large carnivores in Alberta, both the number of elk harvested and hunter success has been increasing throughout the province except in mountain WMUs (Zone 4).

## Management implications

Increasing harvests and abundance of elk indicates that AEP is managing elk sustainably within the province overall. Further, we found that increasing large predator populations do not necessarily equate to a loss in prey populations at the provincial scale. If habitats are sufficient to support a larger prey population, then the prey population should be able to support a larger population of predators [14, 63]. For example, in Alaska as prey populations increased, wolf territory size decreased, leaving more room for additional wolf packs [64]. Nevertheless, a growing elk population might be cause for concern for management of other ungulates. Thus, continued vigilance is required, specifically to protect migration routes for elk into western mountains [47]. Yet, elk in Alberta outside of the mountain units are thriving, and harvest management has been adequate to ensure viable and sustainable herds throughout the province. Moreover, large carnivore populations have increased due to reduced persecution and increased populations of prey [10].

## Acknowledgments

We thank Mariana Nagy-Reis for support and input in the preparation of this manuscript, along with AEP biologists Anne Hubbs and Greg Hale who helped collect and interpret data. We also appreciate many staff, past and present, of Alberta Environment and Parks who collected the elk harvest data. C. Hardie assisted with database management.

## Author Contributions

**Conceptualization:** Mark S. Boyce.

**Formal analysis:** Tyler Trump, Kyle Knopff, Andrea Morehouse, Mark S. Boyce.

**Funding acquisition:** Mark S. Boyce.

**Investigation:** Tyler Trump, Kyle Knopff, Andrea Morehouse, Mark S. Boyce.

**Methodology:** Kyle Knopff, Andrea Morehouse, Mark S. Boyce.

**Project administration:** Mark S. Boyce.

**Resources:** Mark S. Boyce.

**Supervision:** Mark S. Boyce.

**Validation:** Andrea Morehouse, Mark S. Boyce.

**Writing – original draft:** Tyler Trump, Mark S. Boyce.

**Writing – review & editing:** Tyler Trump, Kyle Knopff, Andrea Morehouse, Mark S. Boyce.

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
