## [Decision Letter · Decision Letter 0]

16 Aug 2022

PONE-D-22-14168Sustainable elk harvests in Alberta with increasing predator populationsPLOS ONE

Dear Dr. Boyce,

Thank you for submitting your manuscript to PLOS ONE. After careful consideration, we feel that it has merit but does not fully meet PLOS ONE’s publication criteria as it currently stands. Therefore, we invite you to submit a revised version of the manuscript that addresses the points raised during the review process.

 Can you please address the concerns raised by the expert reviewer?

We look forward to receiving your revised manuscript.

Kind regards,

Avanti Dey, PhD

Staff Editor

PLOS ONE

Journal Requirements:

Reviewers' comments:

Reviewer's Responses to Questions

**Comments to the Author**

1. Is the manuscript technically sound, and do the data support the conclusions?

Reviewer #1: Yes

2. Has the statistical analysis been performed appropriately and rigorously? 

Reviewer #1: Yes

3. Have the authors made all data underlying the findings in their manuscript fully available?

Reviewer #1: Yes

4. Is the manuscript presented in an intelligible fashion and written in standard English?

Reviewer #1: Yes

5. Review Comments to the Author

Reviewer #1: I believe you did a nice job using existing data to address an important management issue that has implications for large mammal management across North America. Throughout large portions of western North America, concerns exist as to the effects of recovering predator populations on ungulate populations. You address this question as it pertains to predator and elk populations in Alberta. I like that you did not focus only on the area(s) where predator effects are hypothesized to be greatest (e.g., Zone 4 in your study), but instead looked at Alberta as a whole to provide a more complete picture. I made a number of edits below that I believe should be addressed prior to publication. Most important, please note my concerns below about the Management Implications section; I think you should remove references to other ungulates and apparent competition as this goes beyond the scope of the paper.

Lines 26, 41: I believe elk scientific name changed to Cervus canadensis

Lines 29-32: I strongly suggest rewording these two sentences so it’s clear you are reporting rates of increase. When I first read the sentence on hunter success, I thought you were saying average hunter success was 0.2% or 0.3% per year (as opposed to increasing by that amount per year). I suggest rewording as: “Over a 26-yr period, average harvest of elk increased by 5.46% per year for unrestricted bull and by 6.64% per year for limited-quota seasons. Also, over the same time frame, average hunter success increased by 0.2% per year for unrestricted bull and by 0.3% per year for limited-quota seasons, but no trend was detected in hunter effort (P>0.05).”

Lines 48-49: Be more specific with this statement or I suggest deleting it. Is this statement specific to Alberta, or Canadian provinces in general? If so, say as much. For many state agencies in the western US, analysis of harvest data is the basis for elk harvest management decisions. These analyses aren’t always published in the peer-reviewed literature, but they happen nonetheless, often by biometricians hired by the agencies to do that very thing. The cited paper (Yoccoz et al 2001) is not sufficient to support your statement (i.e. it doesn’t specifically address big game harvest monitoring systems). Additionally, the paper was published in 2001, and there have been a number of advances in agency big game data monitoring and analysis techniques.

Line 98/Study Area Section: Add description of other ungulates in Alberta, and if possible, relative abundance compared to elk. This would help the reader have some context when interpreting results relative to the predator effect.

Lines 140-144: There’s nothing you can do at this point about the way the data were collected, but we know non-response can lead to bias in harvest data (i.e. non-response bias). As you point out, your real assumption here is that any non-response bias was roughly the same throughout the analysis period. I think that’s reasonable, especially given the alternative is to disregard the entire dataset! With that said, I recommend coming back and discussing your assumption in the Discussion. Specifically, I would emphasize that your results are not likely explained by some corresponding trend in non-response bias through time.

Lines 167-170: I suggest deleting the sentences describing what happened to wolf populations prior to the analysis period. While interesting, it is not germane to the paper and seems misplaced.

Line 241: Delete apostrophe from “Zone’s”

Lines 271-273: You can say rather confidently that elk harvest has increased through the analysis period, but for the most part, you lack strong evidence to say that predator populations have increased over that period. I would reword to make that clear. Your main point still holds: that elk harvests (and almost certainly elk populations) have increased despite recovery of predator populations in recent decades. Also, I assume from your paper that sportsmen are concerned about effects of increasing predator populations on elk. Regardless of what predator populations are exactly doing, your dataset suggests that those concerns are not founded except in Zone 4.

Line 298: Delete “of” after the word, “for”.

Lines 317-330: I recommend deleting most of the management implications section. The references to other ungulates and apparent competition for the first time (in the management implications section) is confusing and misplaced. Are you suggesting that elk and possibly other ungulate populations should be reduced to enhance caribou populations (to reduce apparent competition)? Or are you saying that predator populations should be reduced to enhance caribou? None of this has been mentioned in the paper prior to this point or is even related to the objectives of the paper, and so it leads to confusion by bringing it up here. With that said, as I mentioned earlier, I do believe you should mention in the Study Area section that these other ungulates are present in Alberta, which will provide important context to your analysis.

6. PLOS authors have the option to publish the peer review history of their article (what does this mean?). If published, this will include your full peer review and any attached files.

Reviewer #1: No

---

## [Author Response · Author response to Decision Letter 0]

26 Aug 2022

Response to reviewer 

Thank you for the careful review of our manuscript. We offer the following responses to each comment:

1. We have followed guidelines carefully.

2. All data are freely available on line on the dataverse website, and a doi has been assigned:

doi.org/10.5683/SP3/DINHMO

3. This map in Fig 1 is an original created by coauthor Andrea T. Morehouse and is available for use under the Creative Commons BY 4.0 License. We have inserted an attribution to Dr. Morehouse at the end of the figure caption.

Detailed comments:

Reviewer #1: I believe you did a nice job using existing data to address an important management issue that has implications for large mammal management across North America. Throughout large portions of western North America, concerns exist as to the effects of recovering predator populations on ungulate populations. You address this question as it pertains to predator and elk populations in Alberta. I like that you did not focus only on the area(s) where predator effects are hypothesized to be greatest (e.g., Zone 4 in your study), but instead looked at Alberta as a whole to provide a more complete picture. I made a number of edits below that I believe should be addressed prior to publication. Most important, please note my concerns below about the Management Implications section; I think you should remove references to other ungulates and apparent competition as this goes beyond the scope of the paper.

Thank you for your constructive comments. We agree that reference to apparent competition is beyond the scope of our manuscript and we have deleted this material.

Lines 26, 41: I believe elk scientific name changed to Cervus canadensis

We disagree. The definitive reference for mammal taxonomy is the archived Smithsonian list of mammals and the Mammal species of the World by D. S. Wilson and D. M. Reeder. At the annual meeting of the American Society of Mammalogy I spoke directly with Dr. Wilson about the systematics of elk and he stands by explanation inserted in the Smithsonian list justifying that the latin binomial North American elk remains Cervus elaphus.

Reference: Don E. Wilson & DeeAnn M. Reeder (editors). 2005. Mammal Species of the World. A Taxonomic and Geographic Reference (3rd ed), Johns Hopkins University Press, 2,142 pp.

Lines 29-32: I strongly suggest rewording these two sentences so it’s clear you are reporting rates of increase. When I first read the sentence on hunter success, I thought you were saying average hunter success was 0.2% or 0.3% per year (as opposed to increasing by that amount per year). I suggest rewording as: “Over a 26-yr period, average harvest of elk increased by 5.46% per year for unrestricted bull and by 6.64% per year for limited-quota seasons. Also, over the same time frame, average hunter success increased by 0.2% per year for unrestricted bull and by 0.3% per year for limited-quota seasons, but no trend was detected in hunter effort (P>0.05).”

Thank you. We have accepted the wording change suggested by the reviewer.

Lines 48-49: Be more specific with this statement or I suggest deleting it. Is this statement specific to Alberta, or Canadian provinces in general? If so, say as much. For many state agencies in the western US, analysis of harvest data is [sic] the basis for elk harvest management decisions. These analyses aren’t always published in the peer-reviewed literature, but they happen nonetheless, often by biometricians hired by the agencies to do that very thing. The cited paper (Yoccoz et al 2001) is not sufficient to support your statement (i.e. it doesn’t specifically address big game harvest monitoring systems). Additionally, the paper was published in 2001, and there have been a number of advances in agency big game data monitoring and analysis techniques.

As recommended we have narrowed the scope of our statement to Alberta, and provided an updated reference in which Artelle (2019) conducted a systematic analysis of how science has been used for making wildlife policy decisions in North America. We have removed the Yoccoz et al. (2001) reference. Yes, there are state agencies that employ statisticians, but not most. We believe that our revised statement is correct.

Line 98/Study Area Section: Add description of other ungulates in Alberta, and if possible, relative abundance compared to elk. This would help the reader have some context when interpreting results relative to the predator effect.

Other hunted ungulates are now listed as recommended. We did not list mountain goats or bison because these 2 species are relatively uncommon and not hunted currently.

Lines 140-144: There’s nothing you can do at this point about the way the data were collected, but we know non-response can lead to bias in harvest data (i.e. non-response bias). As you point out, your real assumption here is that any non-response bias was roughly the same throughout the analysis period. I think that’s reasonable, especially given the alternative is to disregard the entire dataset! With that said, I recommend coming back and discussing your assumption in the Discussion. Specifically, I would emphasize that your results are not likely explained by some corresponding trend in non-response bias through time.

Thank you. We fully agree and have inserted a statement about sampling in the Discussion. Fortunately, harvest reporting is now mandatory and non-response bias should disappear as a possible bias.

Lines 167-170: I suggest deleting the sentences describing what happened to wolf populations prior to the analysis period. While interesting, it is not germane to the paper and seems misplaced.

We prefer to keep this material because it provides important context about the huge changes that have happened to large carnivore populations in Alberta during recent years. Wolves were essentially absent from elk ranges only 10 years prior to our analysis and they were still in recovery at the beginning of the period of our data. Similar changes have happened in many other areas of western Canada and USA.

Line 241: Delete apostrophe from “Zone’s”

Thank you, corrected.

Lines 271-273: You can say rather confidently that elk harvest has increased through the analysis period, but for the most part, you lack strong evidence to say that predator populations have increased over that period. I would reword to make that clear. Your main point still holds: that elk harvests (and almost certainly elk populations) have increased despite recovery of predator populations in recent decades. Also, I assume from your paper that sportsmen are concerned about effects of increasing predator populations on elk. Regardless of what predator populations are exactly doing, your dataset suggests that those concerns are not founded except in Zone 4.

The evidence for increasing populations of large carnivores is strong, but otherwise we agree with these comments. The reviewer clearly got our main message and we appreciate the perspective.

Line 298: Delete “of” after the word, “for”.

Thank you. Corrected.

Lines 317-330: I recommend deleting most of the management implications section. The references to other ungulates and apparent competition for the first time (in the management implications section) is confusing and misplaced. Are you suggesting that elk and possibly other ungulate populations should be reduced to enhance caribou populations (to reduce apparent competition)? Or are you saying that predator populations should be reduced to enhance caribou? None of this has been mentioned in the paper prior to this point or is even related to the objectives of the paper, and so it leads to confusion by bringing it up here. With that said, as I mentioned earlier, I do believe you should mention in the Study Area section that these other ungulates are present in Alberta, which will provide important context to your analysis.

As noted above, we agree that the apparent competition issue is beyond the scope of this paper so we have deleted relevant text and references. And as noted above we now list the other ungulates that are hunted in Alberta.

---

## [Editor Report · Decision Letter 1]

14 Sep 2022

Sustainable elk harvests in Alberta with increasing predator populations

PONE-D-22-14168R1

Dear Dr. Boyce,

We’re pleased to inform you that your manuscript has been judged scientifically suitable for publication and will be formally accepted for publication once it meets all outstanding technical requirements.  I participated as a reviewer for the initial evaluation of this manuscript.  

Kind regards,

Chad Bishop

Guest Editor

PLOS ONE
---

## [Editor Report · Acceptance letter]

28 Sep 2022

PONE-D-22-14168R1 

Sustainable elk harvests in Alberta with increasing predator populations 

Dear Dr. Boyce:

I'm pleased to inform you that your manuscript has been deemed suitable for publication in PLOS ONE. Congratulations! Your manuscript is now with our production department. 

Kind regards, 

on behalf of

Dr. Chad Bishop 

Guest Editor

PLOS ONE